# Synbiotic Intervention with an Adlay-Based Prebiotic and Probiotics Improved Diet-Induced Metabolic Disturbance in Mice by Modulation of the Gut Microbiota

**DOI:** 10.3390/nu13093161

**Published:** 2021-09-10

**Authors:** Wei-Chung Chiou, Bei-Hau Chang, Hsiao-Hsuan Tien, Yu-Lin Cai, Ya-Chi Fan, Wei-Jen Chen, Hui-Fang Chu, Yu-Hsin Chen, Cheng Huang

**Affiliations:** 1Department of Biotechnology and Laboratory Science in Medicine, National Yang Ming Chiao Tung University, Taipei 11221, Taiwan; ryanw.chiou@gmail.com (W.-C.C.); bamasaytss520@gmail.com (H.-H.T.); hizamakura999@gmail.com (Y.-L.C.); trana870728@gmail.com (Y.-C.F.); 2Division of Cardiology, Department of Internal Medicine, E-Da Dachang Hospital, Kaohsiung 80794, Taiwan; eagle25@seed.net.tw; 3Syngen Biotech Co., Ltd., Tainan 73055, Taiwan; chen.william@standard.com.tw (W.-J.C.); Chu.HF@syngen.com.tw (H.-F.C.); 4Taichung District Agricultural Research and Extension Station, Council of Agriculture, Changhua 51544, Taiwan; ychen@tdais.gov.tw

**Keywords:** adlay, synbiotic, obesity, insulin resistance, hepatic steatosis, gut microbiota

## Abstract

Metabolic syndrome and its associated conditions, such as obesity and type 2 diabetes mellitus (T2DM), are a major public health issue in modern societies. Dietary interventions, including microbiota-directed foods which effectively modulate the gut microbiome, may influence the regulation of obesity and associated comorbidities. Although research on probiotics and prebiotics has been conducted extensively in recent years, diets with the use of synbiotics remain relatively unexplored. Here, we investigated the effects of a novel synbiotic intervention, consisting of an adlay seed extrusion cooked (ASEC)-based prebiotic and probiotic (*Lactobacillus paracasei* and *Bacillus coagulans*) on metabolic disorders and microbial dysbiosis in high-fat diet (HFD)-induced obese mice. The ASEC-based synbiotic intervention helped improve HFD-induced body weight gain, hyperlipidemia, impaired glucose tolerance, insulin resistance, and inflammation of the adipose and liver tissues. In addition, data from fecal metagenomics indicated that the ASEC-based synbiotic intervention fostered reconstitution of gut bacterial diversity and composition in HFD-induced obese mice. In particular, the ASEC-based synbiotic intervention increased the relative abundance of families *Ruminococcaceae* and *Muribaculaceae* and order Bacteroidales and reduced that of families *Lactobacillaceae*, *Erysipelotrichaceae*, and *Streptococcaceae* in HFD-induced obese mice. Collectively, our results suggest that delayed dietary intervention with the novel ASEC-based synbiotic ameliorates HFD-induced obesity, metabolic disorders, and dysbiosis.

## 1. Introduction

Metabolic syndrome, characterized by abdominal obesity, insulin resistance, hypertension, and hyperlipidemia, increases the onset of cardiovascular diseases (CVD) and type 2 diabetes mellitus (T2DM) and has a prevalence of 20–25% in a meta-analysis of the adult population [1]. The global epidemic of the metabolic syndrome has been shown to be closely associated with the consumption of a high-calorie and low-fiber diet [2]. Microbiota-directed foods (MDFs) have emerged as a strategy to configurate the gut microbiome and to improve the overall metabolic health [3]. The microbiome of the human gastrointestinal (GI) tract houses trillions of microbial cells, among which about 1000 species of anaerobic bacteria are commonly found in each individual [4]. The dietary habits of the host are considered to be important for the composition of the gut microbiota [5]. The gut microbiome participates in many human physiological processes, such as nutrient absorption, metabolism, and immune regulation [6]. Disruption of the gut microbiome has been found to influence the occurrence and development of many diseases, including intestinal inflammation, diabetes, obesity, food allergies, and malignancies [6].

Dietary intervention with probiotics, prebiotics, or synbiotics has been found to restore intestinal homeostasis and may be helpful in resolving metabolic complications in high-fat diet (HFD)-induced obesity [7]. Specifically, supplementation with probiotics alters the composition of the endogenous gut microbiota of the host and, therefore, reshapes the gut microbiome, promoting physiological homeostasis [7]. Meanwhile, prebiotics, such as inulin, lactosucrose, and other non-digestible substances that are selectively metabolized by the beneficial bacteria among the endogenous gut microbiota, have been shown to foster their colonization of the gut microbiome [8]. On the other hand, combinations of probiotics and prebiotics, termed synbiotics, have the advantages of both and improve the abundance of exogenous and endogenous probiotics synergistically, thereby enhancing the resilience of human gut microbial communities [9].

Adlay (*Coix lachryma-jobi* L.), also known as Job’s tears grains or Chinese pearl barley, is a traditional food and herbal medicine in Asia. It is known to have anti-inflammatory, anti-tumor, and anti-allergic properties [10]. Administration of adlay was closely associated with weight loss in HFD-induced obese C57BL/6J mice and resulted in modification of the gut microbiome [11,12]. Meanwhile, polysaccharides from adlay have been reported to be potential prebiotics, improving glucose homeostasis and altering the composition of the gut microbiota in a diabetic mouse model [13]. *Lactobacillus paracasei* and *Bacillus coagulans* (also known as *Lactobacillus sporogenes*) are two probiotics that have been shown to improve metabolic derangements and gut resilience [14,15,16,17], and these have been included in synbiotic formulations [15,18]. However, adlay-based synbiotics remain relatively unexplored in terms of improving diet-induced metabolic disorders and reshaping the gut microbiota.

In this study, delaying dietary intervention until eight weeks after the start of HFD feeding was used to simulate the clinical situation of obesity, in which metabolic dysfunction has already developed. We aimed to determine whether intervention with adlay seed extrusion cooked (ASEC)-based prebiotic (the ASEC group), probiotics (*Lactobacillus paracasei* and *Bacillus coagulans*; the PRO group), and the synbiotic (the ASEC + PRO group) ameliorates diet-induced obesity and metabolic disorders and modulates the abundance and diversity of the gut microbiota.

## 2. Materials and Methods

### 2.1. Animals and Diets

Male C57BL/6J mice at four weeks of age were purchased from the National Laboratory Animal Center, Taipei, Taiwan. Mice were housed in a temperature-controlled room with a 12 h light/dark cycle, in the Animal Center of the National Yang Ming Chiao Tung University, Taipei, Taiwan (IACUC no. 1080320), and all procedures followed The Guide for the Care and Use of Laboratory Animals (NIH publication, 85-23, revised 1996) and the guidelines of the Animal Welfare Act, Taiwan. At five weeks of age, mice were randomly divided into two groups, where one continued with the normal diet (ND; *n* = 8), the other changed to the HFD (*n =* 32) and were fed for another eight weeks before receiving a 12-week intervention. Diet-induced obese mice were divided into four groups: the HFD group (HFD, *n =* 8), HFD-ASEC group (ASEC, *n =* 8), HFD + probiotics (PRO, *n =* 8), and HFD-ASEC + probiotics (ASEC + PRO, *n =* 8). The main components of ASEC (*Coix lacryma-jobi* L. var. ma-yuen Stapf) from the Taichung District Agricultural Research and Extension Station (Changhua, Taiwan) were published previously [19]. The specific compositions of the experimental diets are listed in Table 1. Commercially available probiotics, *Lactobacillus paracasei* LCW-23 (AU2019100286) and *Bacillus coagulans* BACO-17 (AU2020103929) (Syngen Biotech Co., Ltd., Taiwan), were stored at 4 °C before use and were administered at 5 × 10^7^ CFU/species/day via oral gavage. Casein (Cat. 901293), L-cystine (Cat. 101454), corn starch (Cat. 902956), dextrinized corn starch (Cat. 960429), sucrose (Cat. 199631), lard (Cat. 902140), cellulose (Cat. 900453), AIN-93G mineral mix (Cat. 960400), AIN-93 vitamin mix (Cat. 960402), and choline bitartrate (Cat. 101384) were purchased from MP Biomedicals (Santa Ana, CA, USA). Soybean oil (Cat. 7032) was from Taiwan Sugar Corp.

### 2.2. Biochemical Characterization

The levels of serum triglyceride (TG, Cat. 15809671), total cholesterol (TC, Cat. 15809669), high-density lipoprotein cholesterol (HDL-C, Cat. 15809736), glutamate oxaloacetate aminotransferase (GOT, Cat. 15809542), glutamate pyruvate aminotransferase (GPT, Cat. 15809554), total bilirubin (TBIL, Cat. 15809657), creatinine (CRE, Cat. 15809475), blood urea nitrogen (BUN, Cat. 15809425), and amylase (AMYL, Cat. 16015150) were measured using a FUJI DRI-CHEM NX500 analyzer (Fujifilm, Japan). The LDL cholesterol (LDL-C) concentration was calculated from the Friedelwald equation: TC − (HDLC + TG/5) [20]. The levels of resistin, leptin, plasminogen activator inhibitor-1 (PAI-1), and ghrelin were determined using Luminex multiplex systems provided by Yu-Shing Bio-Tech Co., Taiwan.

### 2.3. Histological Analysis of Liver and eWAT

Following the experimental procedure published previously [21], the liver and epididymal white adipose tissue (eWAT) from mice were weighed and fixed in 10% neutral buffered formalin (NBF, Cat. HT501640, Sigma-Aldrich, St. Louis, MO, USA). Hematoxylin and eosin (H&E) staining was performed, and all specimens were observed at 200× magnification under a BIO-2 T bright-filed microscope (BEL^®^ Engineering, Monza, Italy).

### 2.4. Blood Glucose, Serum Insulin, Homeostasis Model Assessment of Insulin Resistance Index (HOMA-IR), and Intraperitoneal Glucose Tolerance Tests (IPGTT) 

Determination of fasting blood glucose, insulin, HOMA-IR, and glucose tolerance followed the procedures published previously [22]. Briefly, the glucose levels of mice were determined by a glucose analyzer (EasyTouch ET-201 GU Blood Glucose Uric Acid meter, Taiwan). Insulin Ultra-Sensitive kits (Cat. 62IN2PEG) from Cisbio, Bedford, MA, USA were used to determine serum insulin levels. HOMA-IR was calculated as [fasting insulin (µU/mL) × fasting glucose (mg/dL) × 0.05551]/22.5. Glucose tolerance was determined using IPGTT, in which glucose was injected at a concentration of 1.0 g/kg body weight and the glucose levels were measured by tail vein sampling.

### 2.5. Quantitative Real-Time PCR

Quantitative real-time PCR (qPCR) was performed according to a previous report [23]. Briefly, total mRNA was extracted from the tissues using Invitrogen™ TRIzol™ Reagent (Cat. 15596018, Thermo Fisher Scientific, Waltham, MA, USA) and chloroform (Cat. 1.02445.1000, Merck, Germany). Then, the extracted mRNA was reverse-transcribed into cDNA using the RevertAid™ First Strand cDNA Synthesis Kit (Cat. K1622, Thermo Fisher Scientific, USA). qPCR was performed with Applied Biosystems™ SYBR™ Green PCR Master Mix (Cat. 4309155, Thermo Fisher Scientific, USA). The relative mRNA expression of genes of interest was calculated with the ΔΔCt method and was normalized to that of GAPDH. Primers used in qPCR are listed in Appendix A and are synthesized by Genomics (Taiwan).

### 2.6. Gut Microbiota Analysis

To analyze the gut bacterial composition, fecal metagenomics was performed following the procedure described previously [24]. Briefly, QIAamp DNA Stool Mini Kit (Cat. 51504, Qiagen, Germany) was used to extract fecal genomic DNA, and the V3–V4 region of the 16S rRNA was amplified before adding multiplexing indices and Illumina sequencing adapters. For bioinformatics analysis, all effective reads were analyzed as published previously [25]. 

### 2.7. Statistical Analysis

Cohen’s d test was used to determine the sample size (d = 1.4; alpha error = 0.05; statistical power = 80%). GraphPad Prism 7.0 software (San Diego, CA, USA) was used to perform statistical analyses. One-way analysis of variance (ANOVA) with Tukey’s post-hoc tests were performed to determine the statistical significance (*p* < 0.05) between groups, indicated by different letters (e.g., a, b). Kruskal-Wallis with Dunn’s post-hoc tests were used in the statistical analysis of OTUs from fecal metagenomics, where the statistical significance was set at *p* < 0.05.

## 3. Results

### 3.1. Effects of an ASEC-Based Synbiotic Diet on Body Weight Gain, Adipose Tissue Hypertrophy, and Dyslipidemia in HFD-Induced Obese Mice

The experimental scheme of this study is illustrated in Figure 1A. Ingredients of experimental diets are listed in Table 1. Specifically, ND, HFD, and HFD-ASEC provide 3.8, 4.8, and 4.8 kcal/g, respectively; meanwhile, about 45% energy of HFD and HFD-ASEC is from fat. Body weight changes of all groups were recorded weekly, as shown in Figure 1B. The delayed intervention with ASEC (*p* = 0.0003), PRO (*p* < 0.0001), or ASEC + PRO (*p* < 0.0001) prevented the progression of body weight gain compared to the HFD group. In particular, the body weight of the ASEC + PRO group had a moderate decrease compared to the ASEC and PRO groups. HFD feeding resulted in obvious adipocyte hypertrophy compared to the ND group (*p* < 0.0001), and the delayed intervention with ASEC (*p* < 0.0001), PRO (*p* < 0.0001), or ASEC + PRO (*p* < 0.0001) reversed the diet-induced adipocyte hypertrophy (Figure 1C,E). As shown in Figure 1D, the average eWAT weight of the ND, HFD, ASEC, PRO, and ASEC + PRO groups were 0.76, 1.96, 1.43, 1.48, and 1.11 g, respectively. Interestingly, fat deposition in the eWAT was effectively alleviated in the ASEC + PRO group (*p* = 0.0013), resulting in an eWAT weight comparable to that of the ND group (*p* = 0.4095). As shown in Figure 1F, HFD feeding increased the serum TG level to over 30 mg/dL; meanwhile, all intervention groups (ASEC, *p* < 0.0001; PRO, *p* < 0.0001; ASEC + PRO, *p* = 0.0003) potently reduced the serum TG concentration, down to a level close to that of the ND group. In Figure 1G, HFD feeding raised the serum TC level to about 300 mg/dL, up from about 100 mg/dL in the ND group, and intervention with ASEC (*p* = 0.0181), PRO (*p* < 0.0001), or ASEC + PRO (*p* < 0.0001) decreased the serum TC level substantially in HFD-induced obese mice. In particular, the delayed intervention with PRO (*p* = 0.9285) or ASEC + PRO (*p* > 0.9999) was found to return the level of serum TC close to that of the ND group. The HDL-C levels of all groups did not differ significantly, as shown in Figure 1H. As for the LDL-C level (Figure 1I), a tremendous increase was seen in the HFD group (*p* < 0.0001) compared to the ND group, while this phenomenon was reversed in the PRO (*p* < 0.0001) and ASEC + PRO (*p* < 0.0001) groups.

### 3.2. Effects of an ASEC-Based Synbiotic Diet on Glucose Tolerance and Insulin Resistance in HFD-Induced Obese Mice

Records of fasting blood glucose are plotted in Figure 2A. An ascending profile was seen after feeding an HFD (*p* < 0.0001), while this trend was averted in the ASEC (*p* < 0.0001), PRO (*p* = 0.0039), and ASEC + PRO (*p* = 0.0013) groups. Consistent with the profile of fasting blood glucose, impaired glucose tolerance was seen only in the HFD group (Figure 2B). HFD feeding induced an increase in the fasting insulin level by over 2.5-fold (*p* < 0.0001) compared to the ND group (Figure 2C). Meanwhile, the fasting insulin level was significantly reduced in all intervention groups (ASEC, *p* < 0.0001; PRO, *p* < 0.0001; ASEC + PRO, *p* < 0.0001). The HOMA-IR index of the HFD group was about three-fold greater (*p* < 0.0001) than the other groups (Figure 2D). Further, the levels of key biomarkers of metabolic syndrome, leptin, resistin, adiponectin, PAI-1, and ghrelin were measured. As shown in Figure 2E,F, increases in the levels of serum leptin (*p* < 0.0001) and resistin (*p* < 0.0001) were seen after feeding an HFD, and this was prevented in all intervention groups. On the other hand, the levels of adiponectin (Figure 2G), PAI-1 (Figure 2H), and ghrelin (Figure 2I) did not differ significantly between the groups.

### 3.3. Effects of an ASEC-Based Synbiotic Diet on Hepatic Steatosis in HFD-Induced Obese Mice

To investigate the development of hepatic steatosis, the lipid content and the morphology of the liver were examined. The liver weight gain from HFD feeding (*p* < 0.0001) was precluded when the ASEC, PRO, or ASEC + PRO intervention was made (Figure 3A). As for the lipid content in the liver, liver TG (Figure 3B; *p* = 0.0462) and liver TC (Figure 3C; *p* < 0.0001) were found to increase after HFD feeding, while this outcome was improved in all intervention groups. In particular, the liver TC (*p* = 0.0554) and TG (*p* > 0.9999) levels of the ASEC + PRO group were comparable to those of the ND group. Corresponding to the measurements of the lipid content of the liver, compared to the HFD group, a visible decrease in lipid droplets in the liver biopsy (Figure 3D) was observed in all intervention groups. Next, the indicators of hepatic injury, GOT, GPT, and TBIL were examined. As shown in Figure 3E–G, hepatic injury resulting from HFD feeding was improved in the ASEC, PRO, and ASEC + PRO groups. The levels of creatinine (CRE), blood urea nitrogen (BUN), and amylase (AMYL) were measured to determine whether the respective functions of the kidneys and pancreas were affected by HFD feeding, as obesity and metabolic disorders induce systemic low-grade inflammation, including the kidney and pancreas [26,27]. As shown in Figure 3H,I, HFD-induced kidney dysfunction was ameliorated by intervention with ASEC, PRO, or ASEC + PRO. As for the level of serum amylase (Figure 3J), that of the ND group was not statistically different from other groups.

### 3.4. Effects of an ASEC-Based Synbiotic Diet on Proinflammatory Factors in the Liver and Adipose Tissues in HFD-Induced Obese Mice

Obesity-related metabolic disorders elicit systemic low-grade inflammation by increasing the levels of proinflammatory factors, such as monocyte chemoattractant protein-1 (MCP-1), CD11c, tumor necrosis factor-α (TNF-α), leptin, lipopolysaccharide binding protein (LBP), and interleukin-6 (IL-6) [26,28,29]. Feeding an HFD increased the mRNA expression of MCP-1 (*p* < 0.0001), TNF-α (*p* = 0.0004), IL-6 (*p* < 0.0001), LBP (*p* < 0.0001), leptin (*p* < 0.0001), and CD11c (*p* = 0.0001) in the eWAT (Figure 4A–F), and MCP-1 (*p* < 0.0001), TNF-α (*p* = 0.0075), IL-6 (*p* < 0.0001), and LBP (*p* = 0.0054) in liver tissues (Figure 4G–J). All Intervention groups mitigated the upregulated transcriptional level of proinflammatory factors induced by HFD feeding. Interestingly, the reduction in the eWAT TNF-α mRNA level in the PRO group (*p* = 0.0930) was relatively moderate compared to the ASEC (*p* = 0.0007) and ASEC + PRO (*p* = 0.0005) groups. In addition, the mRNA expression level of liver LBP was significantly downregulated after ASEC (*p* < 0.0001), PRO (*p* < 0.0001), or ASEC + PRO (*p* < 0.0001) intervention compared to the ND group.

### 3.5. Effects of an ASEC-Based Synbiotic Diet on the Diversity of Gut Microbiota and Gut Bacteroidetes in HFD-Induced Obese Mice

Using the variable region V3–V4 of the 16S rRNA as the operational taxonomic unit (OTU), the diversity of the composition of gut bacteria between the groups was characterized using principal coordinates analysis (PCoA) (Figure 5A). HFD feeding led to a distinct clustering of gut microbiota composition in PCoA, having the greatest distance to the ND group. Besides, the variance of the HFD group was greater than that of the ND group. Partly overlapping with the HFD group, the variance of the PRO group was slightly reduced. As for the ASEC and ASEC + PRO groups, they were of greater similarity to the ND group compared with the HFD and PRO groups. In Figure 5B, a dendrogram of entities showed that each group clustered in close proximity. Furthermore, the Chao1 index and the Simpsons diversity index were calculated to determine the richness (Figure 5C) and community composition (Figure 5D) of the gut microbiome, respectively. A reduction in the bacterial richness and community diversity was observed after feeding an HFD. Interestingly, although all the dietary intervention groups have improved community diversity compared with the HFD group, the bacterial richness of the ASEC group also increased significantly (*p* = 0.0311). The ratio of Firmicutes to Bacteroidetes (F/B ratio) positively correlates with the extent of obesity [30]. Here, the relative abundance of Firmicutes (Figure 5E) increased after feeding an HFD compared to the ND group (*p* < 0.0001); meanwhile, the PRO (*p* = 0.0002) or ASEC + PRO (*p* = 0.0009) intervention reduced the relative abundance of Firmicutes. On the other hand, the relative abundance of Bacteroidetes (Figure 5F) decreased markedly after HFD feeding compared to the control group (*p* < 0.0001), while the ASEC + PRO intervention significantly increased the abundance of Bacteroidetes (*p* < 0.0001) in HFD-induced obese mice. In addition, the delayed intervention with ASEC, PRO, or ASEC + PRO helped shift the F/B ratio toward that of the ND group (Figure 5G). Regarding the composition of gut bacteria (Figure 5H), expansion of phyla Bacteroidetes, Actinobacteria, and Deferribacteres was seen in the ASEC + PRO group; meanwhile, the ASEC intervention also increased the population of Actinobacteria and Deferribacteres. 

### 3.6. Effects of an ASEC-Based Synbiotic Diet on Bacterial Dysbiosis in HFD-Induced Obese Mice

In the overall analysis of the richness of the gut microbiota, 134 OTUs from 21 families, including three OTUs without defined families, differed significantly after intervention with ASEC, PRO, or ASEC + PRO, compared to the HFD group. As shown in Figure 6A, with respect to the HFD group, 46 OTUs (15 families) in the ND group, 75 OTUs (15 families) in the ASEC group, 33 OTUs (11 families) in the PRO group, and 67 OTUs (13 families) in the ASEC + PRO group had increased markedly, while 21 OTUs (8 families) in the ND group, 9 OTUs (5 families) in the ASEC group, 15 OTUs (5 families) in the PRO group, and 13 OTUs (8 families) in the ASEC + PRO group were substantially reduced. Collectively, 27 OTUs (10 families; 2 OTUs without defined families) in the ASEC group, 12 OTUs (6 families) in the PRO group, and 26 OTUs (8 families) in the ASEC + PRO group were restored to levels close to the ND group. Moreover, compared to PRO intervention, the number of OTUs improved by the ASEC or ASEC + PRO intervention doubled in HFD-induced obese mice. In particular, the expansion of family *Muribaculaceae* was more pronounced in the ASEC + PRO group (*p* = 0.0177) than in the ASEC (*p* > 0.9999) and PRO groups (*p* > 0.9999).

Further, bacteria of the main phyla in the gut microbiome were investigated with OTUs. After HFD feeding, the populations of phylum Actinobacteria (Figure 6B; *p* = 0.0003) and families *Lachnospiraceae* (Figure 6C; *p* = 0.0235), *Lactobacillaceae* (Figure 6D; *p* < 0.0001), *Erysipelotrichaceae* (Figure 6E; *p* = 0.0005), and *Streptococcaceae* (Figure 6F; *p* < 0.0001) in phylum Firmicutes were found to increase. Meanwhile, family *Ruminococcaceae* (Figure 6G; *p* = 0.0058) in phylum Firmicutes and families *Muribaculaceae* (formerly known as S24-7; Figure 6H; *p* < 0.0001) and *Bacteroidaceae* (Figure 6I; *p* < 0.0001) in phylum Bacteroidetes were reduced. These families constituted about 90% of gut bacteria in HFD-induced obese mice. Specifically, the ASEC intervention alone helped restore the relative abundance of families *Lactobacillaceae* (*p* = 0.0289), *Erysipelotrichaceae* (*p* = 0.3778), *Streptococcaceae* (*p* = 0.0560), *Ruminococcaceae* (*p* = 0.0229), and *Bacteroidaceae* (*p* = 0.0539) in HFD-induced obese mice. As for the PRO group, a similar trend of improvements on HFD-induced dysbiosis was seen, except for families *Streptococcaceae* (*p* > 0.9999) and *Ruminococcaceae* (*p* > 0.9999). Interestingly, the relative abundance of families *Ruminococcaceae* (*p* = 0.0678) and *Muribaculaceae* (*p* = 0.0177) and order Bacteroidales (*p* = 0.0072) was raised in the ASEC + PRO group compared with the HFD group. Regarding probiotics supplementation, the population of *Bacillus coagulans* was significantly raised in the PRO (*p* = 0.0001) and ASEC + PRO (*p* = 0.0023) groups compared with the other groups. Although *Lactobacillus paracasei* was undetermined, the relative abundance of *Lactobacillaceae* was slightly increased in the PRO and ASEC + PRO groups.

## 4. Discussion

The high prevalence of obesity and metabolic syndrome is a particular worldwide health problem at present. Closely associated with the overall metabolic health, diet composition has the ability to modulate the gut microbiome [31]. Studies have shown the importance of the gut microbiota in the development of metabolic disorders, diabetes, and cardiovascular diseases, and in the health of the nervous system [5,32,33]. Based on the knowledge of traditional Asian medicine and the findings of current studies, adlay has been proven to have anti-inflammatory, anticancer, and anti-diabetic properties [10,13,34]. In addition, the prebiotic potential of adlay has been discussed [11]. Foods that promote the growth of probiotics are regarded as prebiotics, and these often contain oligosaccharides and polysaccharides [8,13,35]. Given that the polysaccharide content of adlay seed is about 0.7% (g/g), we investigated whether incorporating adlay in the synbiotic formulation containing *Lactobacillus paracasei* and *Bacillus coagulans* improves metabolic disorders in HFD-induced obese mice.

As a risk factor for insulin resistance and type 2 diabetes, systemic low-grade inflammation is partly the result of obesity-related metabolic endotoxemia, as an elevated level of circulating lipopolysaccharides (LPS) resulting from increased gut permeability leads to inflammation of the LPS-infiltrated adipose tissues and liver [36,37]. Specifically, the infiltrated LPS is transferred to the receptor protein CD14 in the plasma membrane with the help of LBP, and the complex generated activates the transcription of cytokines, such as TNF-α and IL-6, by upregulating Toll-like receptor 4 (TLR4)-mediated c-Jun N-terminal kinase (JNK) and nuclear factor-kappa B (NF-κB) signaling pathways [38]. In support of this rationale, our data show that HFD feeding instigated inflammation in adipose tissues and the liver, with rising mRNA levels of LBP, proinflammatory factors (TNF-α IL-6, and MCP-1), and an inflammation-related transmembrane protein (CD11c), implying the development of a leaky gut. However, this inference has to be further validated by the plasma LPS level and the expression levels of TLR4 and CD14 in intestinal epithelial cells. In parallel with increased visceral adiposity, the rise in circulating leptin and resistin levels exacerbates the local inflammation of adipose tissues, leading to obesity-related insulin resistance [39,40]. Moreover, the expression levels of tight junction proteins, including occludins, claudins, and zonula occludens [41], are worth investigating given the close association between the loss of intestinal epithelial barrier integrity and the onset of metabolic disorders to confirm the effects of adlay-based synbiotic treatment on colonic segment integrity.

As for the gut bacterial composition, HFD-induced dysbiosis disrupts the integrity of the gut barrier [37], reduces the fundamental population of immunosuppressive LPS producers in order Bacteroidales (phylum Bacteroidetes) [42], and also reduces the relative abundance of families *Lachnospiraceae* and *Ruminococcaceae* (phylum Firmicutes), the two families highly rich in short-chain fatty acid (SCFA)-producing bacteria [43]. It is known that SCFAs enhance intestinal barrier function by regulating the expression of tight junction proteins [43,44]. In HFD-induced obese mice, the ASEC, PRO, or ASEC + PRO intervention not only allowed immunosuppressive LPS producers to flourish but also stimulated the expansion of families containing SCFA-producing bacteria. Indeed, the promotion of these bacteria indirectly helped rebalance the integrity of the gut barrier and the immunogenic/immunosuppressive LPS ratio [42,44]. Thus, the downregulated mRNA level of LBP in the liver and adipose tissues after the ASEC, PRO, or ASEC + PRO intervention may be closely associated with the reduced immunogenic LPS levels in the circulation. Meanwhile, the relative abundance of families *Streptococcaceae* and *Erysipelotrichaceae*, which have been reported to be associated with nonalcoholic fatty liver disease (NAFLD) progression, increased in HFD-induced obese mice, and the relative abundances of the family *Muribaculaceae*, which has been reported to correlate with the fecal propionate concentration [45], reduced by more than three-fold in the HFD group. Therefore, it will be interesting to further investigate the effects of ASEC-based symbiotic on SCFAs concentrations. 

Here, we demonstrated that feeding an HFD for eight weeks successfully induced obesity in mice, and the delayed intervention with ASEC, PRO, or ASEC + PRO mitigated the increase in the body weight, adipose tissue hypertrophy, and dyslipidemia. Moreover, delayed intervention with ASEC, PRO, or ASEC + PRO reversed HFD-induced impaired glucose tolerance, insulin resistance, and elevated serum leptin and resistin levels. Regarding the liver and kidney function, delayed intervention with ASEC, PRO, or ASEC + PRO reduced lipid accumulation in the liver and improved hepatic inflammation and kidney function in HFD-induced obese mice. Consistently, the increase in mRNA levels of proinflammatory factors in the liver and eWAT was markedly reduced after ASEC or ASEC + PRO intervention. Regarding the gut microbiome, the ASEC + PRO group not only helped ameliorate HFD-induced dysbiosis but also had a relatively consistent profile across the diversity, richness, and gut microbiota composition compared with the ASEC and PRO groups. Rodent models of diet-induced obesity have been widely used to investigate human metabolic perturbations [46]. Considering that the effects of ASEC-based synbiotic on metabolic health and gut microbiota were based on a sample size of eight per group in a single mouse strain, this research should be regarded as a pilot study that needs to be further validated in experiments of a larger sample size to help extrapolate to clinical therapeutics.

## 5. Conclusions

In conclusion, we showed that HFD feeding leads to body weight gain, adipocyte hypertrophy in the eWAT, dyslipidemia, impaired glucose tolerance, insulin resistance, lipid accumulation in the liver, hepatic and eWAT inflammation, and the loss of overall gut bacterial diversity. Delayed intervention with ASEC, PRO, or ASEC + PRO helped ameliorate metabolic disorders, organ dysfunction, and other obesity-related pathophysiological indices in HFD-induced obese mice. Regarding HFD-induced dysbiosis, the ASEC intervention alone helped restore families *Lactobacillaceae*, *Erysipelotrichaceae*, *Streptococcaceae*, *Ruminococcaceae*, and *Bacteroidaceae* to levels comparable to the ND group, and a synergistic effect of ASEC + PRO on the relative abundances of family *Muribaculaceae* and order Bacteroidales was observed. Thus, owing to the extent of gut bacterial modulation, the positive effects of ASEC + PRO on HFD-induced metabolic disorders and dysbiosis highlighted the potential of the ASEC-based synbiotic for the treatment of diet-induced obesity. The results of our study suggest that synbiotic intervention with an adlay-based prebiotic and probiotic ameliorates dietary fat-induced obesity and gut microbiota dysbiosis and provides a potential dietary strategy to overcome metabolic syndrome.

## Figures and Tables

**Figure 1 nutrients-13-03161-f001:**
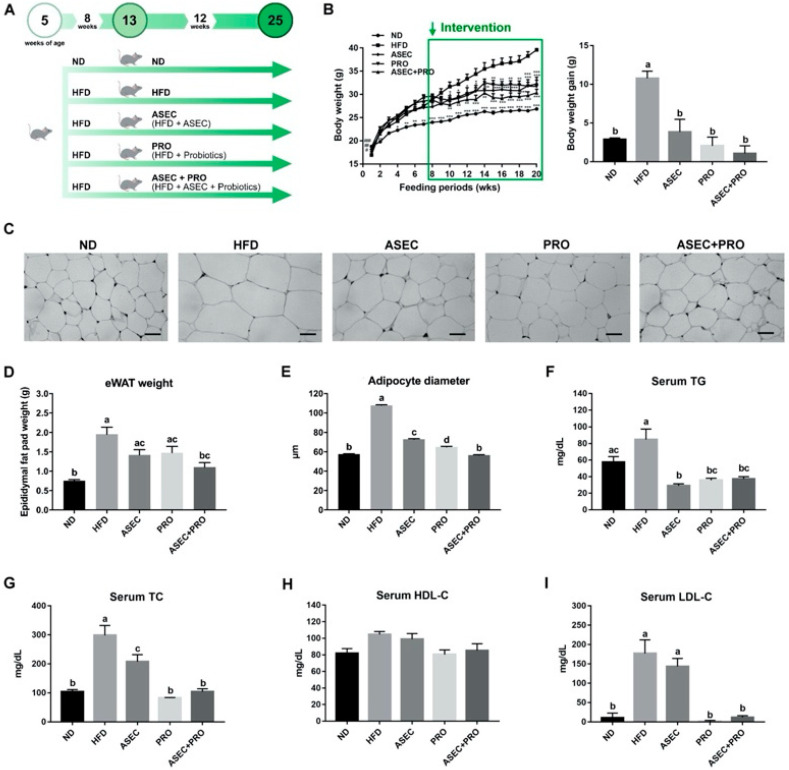
Intervention with ASEC, PRO, or ASEC + PRO improved body weight gain, fat deposition, and dyslipidemia in HFD-induced obese mice. (**A**) A graphical illustration of the treatment scheme. Male C57BL/6J 5-week-old mice were fed a normal diet (ND) or a high-fat diet (HFD) for 8 weeks before receiving the ASEC, PRO, or ASEC + PRO intervention. (**B**) Changes in body weight over time and the body weight gain at the end time point. The body weight gain was the weight difference between the beginning and the end of the experiment. (**C**) H&E staining of adipocytes in the epididymal white adipose tissue (eWAT). The scale bar indicates 100 µm. (**D**) The weight of eWAT. (**E**) The diameter of adipocytes. (**F**) Serum triglyceride (TG). (**G**) Serum total cholesterol (TC). (**H**) Serum high-density lipoprotein cholesterol (HDL-C). (**I**) Serum low-density lipoprotein cholesterol (LDL-C). Data (*n =* 8) are expressed as the mean ± SEM. ASEC, adlay seed extrusion-cooked prebiotic; PRO, probiotics. Statistical significance (*p* < 0.05) is indicated by different letters (e.g., a, b). * *p* < 0.05, ** *p* < 0.01, *** *p* < 0.001.

**Figure 2 nutrients-13-03161-f002:**
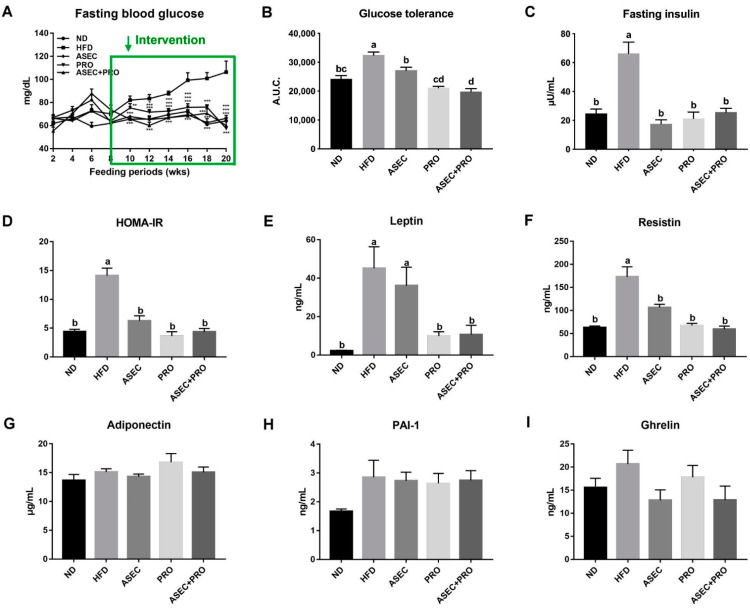
Intervention with ASEC, PRO, or ASEC + PRO improved impaired glucose intolerance, insulin resistance, and related pathophysiological indicators in HFD-induced obese mice. (**A**) Fasting blood glucose. Fasting blood glucose was measured by a glucose analyzer biweekly. (**B**) Assessment of glucose tolerance using intraperitoneal glucose tolerance tests (IPGTTs). Glucose was intraperitoneally injected at a concentration of 1.0 g/kg body weight and the glucose levels were measured by tail vein sampling. (**C**) Fasting insulin levels. Insulin Ultra-Sensitive kits were used for measurement. (**D**) The homeostasis model assessment of insulin resistance index (HOMA-IR). HOMA-IR was calculated as [fasting insulin (µU/mL) × fasting glucose (mg/dL) × 0.05551]/22.5. Levels of serum leptin (**E**), resistin (**F**), adiponectin (**G**), PAI-1 (**H**), and ghrelin (**I**) were determined by Luminex multiplex systems. Data (*n =* 8) are expressed as the mean ± SEM. Statistical significance (*p* < 0.05) is indicated by different letters (e.g., a, b). * *p* < 0.05, ** *p* < 0.01, *** *p* < 0.001.

**Figure 3 nutrients-13-03161-f003:**
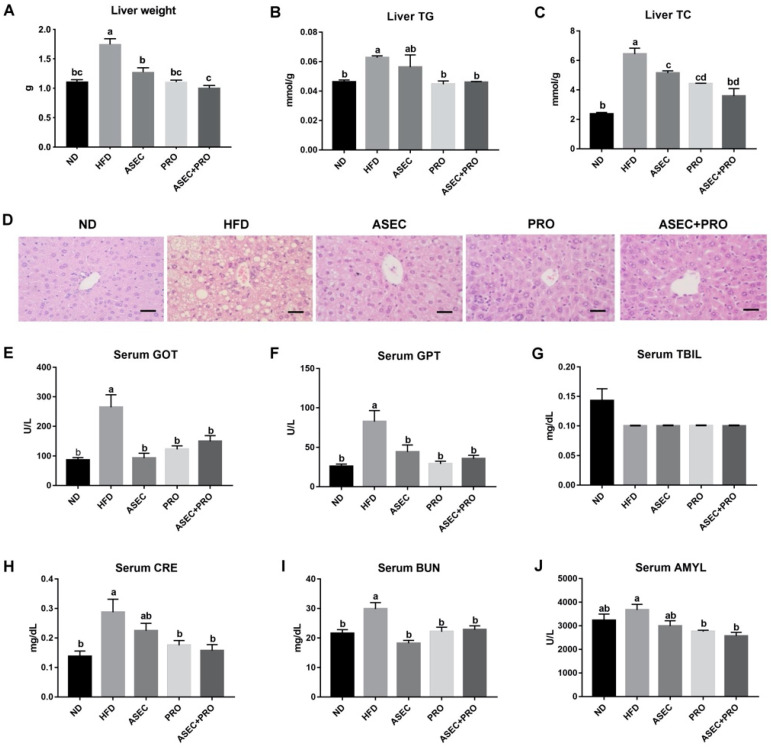
Intervention with ASEC, PRO, or ASEC + PRO alleviated hepatic steatosis. (**A**) Liver weight. (**B**) Liver TG. (**C**) Liver TC. (**D**) H&E staining of the liver biopsy. The scale bar is 100 µm. (**E**) Serum glutamate oxaloacetate aminotransferase (GOT). (**F**) Serum glutamate pyruvate aminotransferase (GPT). (**G**) Serum total bilirubin (TBIL). (**H**) Serum creatinine (CRE). (**I**) Serum blood urea nitrogen (BUN). (**J**) Serum amylase (AMYL). Biochemical characterization was performed using FUJI SRI-CHEM slides and a FUJI DRI-CHEM analyzer. Data (*n =* 8) are expressed as the mean ± SEM. Statistical significance (*p* < 0.05) is indicated by different letters (e.g., a, b).

**Figure 4 nutrients-13-03161-f004:**
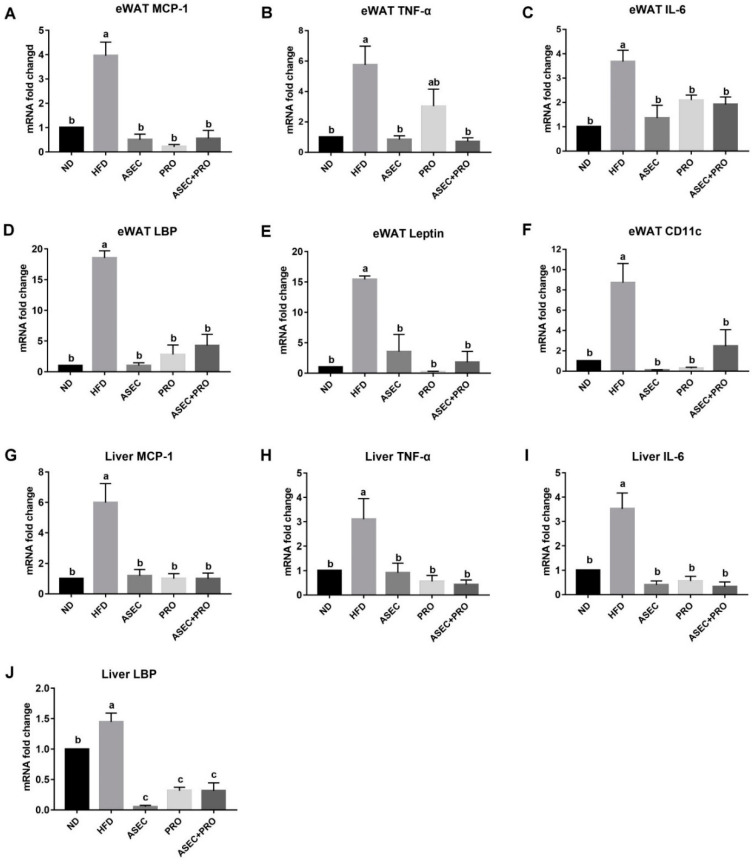
Intervention with ASEC, PRO, or ASEC + PRO reduced the transcription level of inflammatory genes in the liver and adipose tissues in HFD-induced obese mice. (**A**–**F**) The mRNA expression levels of MCP-1 (**A**), TNF-α (**B**), IL-6 (**C**), LBP (**D**), leptin (**E**), and CD11c (**F**) in the epididymal white adipose tissue (eWAT). (**G**–**J**) The mRNA expression levels of MCP-1 (**G**), TNF-α (**H**), IL-6 (**I**), and LBP (**J**) in the liver. Quantitative real-time PCR (qPCR) was performed. Data (*n =* 8) are expressed as the mean ± SEM. MCP-1, monocyte chemoattractant protein-1; TNF-α, tumor necrosis factor-α; IL-6, interleukin-6; LBP, lipopolysaccharide binding protein. Statistical significance (*p* < 0.05) is indicated by different letters (e.g., a, b).

**Figure 5 nutrients-13-03161-f005:**
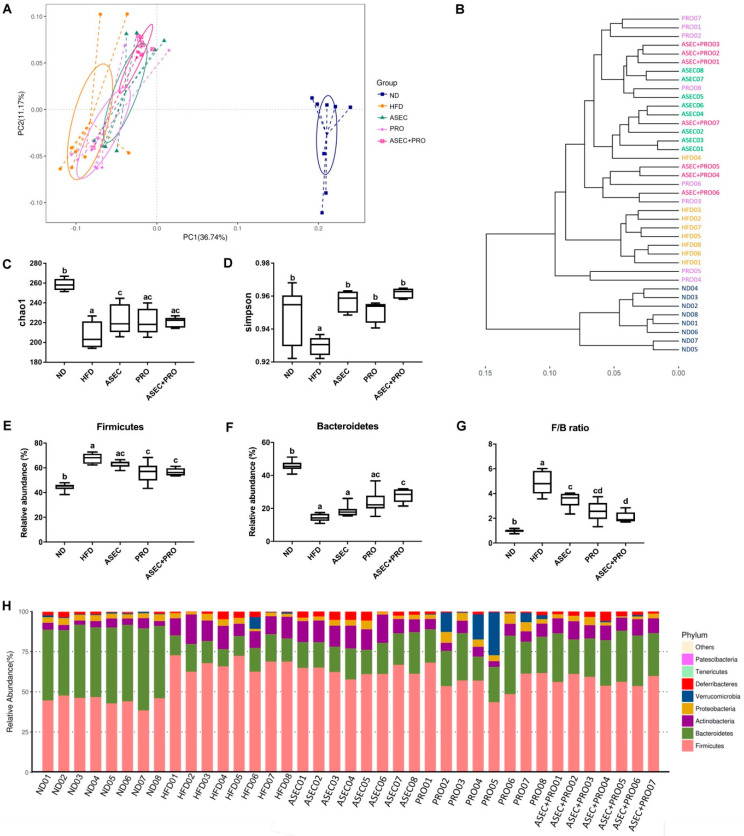
Intervention with ASEC, PRO, or ASEC + PRO improved the overall diversity and structure of gut bacteria in HFD-induced obese mice. Fecal metagenomics was performed using the V3–V4 of the 16S rRNA as the operational taxonomic unit (OTU). (**A**) Principal coordinates analysis (PCoA) of gut microbiota. (**B**) Unweighted pair group method with arithmetic mean (UPGMA) tree of gut bacteria. (**C**) Bacterial richness using the Chao1 index. (**D**) Bacterial diversity using the Simpsons diversity index. The relative abundance of phylum Firmicutes (**E**) and phylum Bacteroidetes (**F**) is shown. (**G**) The ratio of Firmicutes to Bacteroidetes (F/B ratio). (**H**) Taxonomic profiling of gut bacteria at the phylum level. Data (*n =* 7–8) are expressed as the mean ± SEM. Statistical significance (*p* < 0.05) is indicated by different letters (e.g., a, b).

**Figure 6 nutrients-13-03161-f006:**
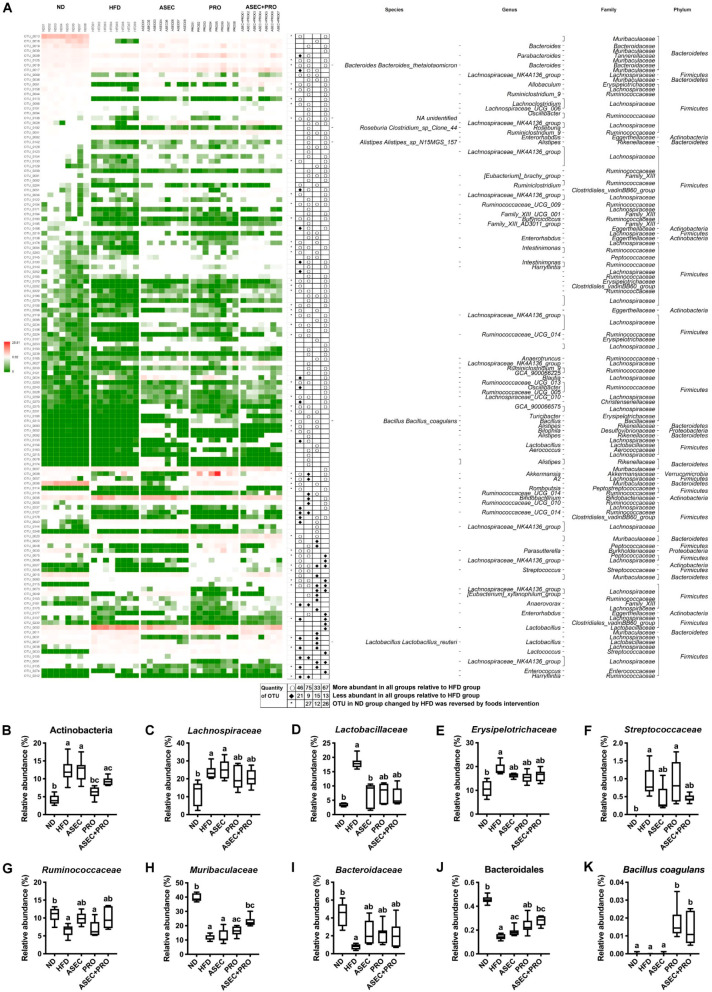
Intervention with ASEC, PRO, or ASEC + PRO helped reconstitute HFD-induced dysbiosis. (**A**) The heatmap of the top 134 OTUs. Compared to the HFD group, hollow circles and black diamonds indicate the increases and decreases in the relative abundances of OTUs, respectively, in the ND, ASEC, PRO, and ASEC + PRO groups. Black asterisks indicate that OTUs altered by HFD were reversed to levels close to the ND group after intervention with ASEC, PRO, or ASEC + PRO. The taxonomy (phylum, family, genus, and species) of each OTU is shown on the right of the heatmap for reference. (**B**) Phylum Actinobacteria. (**C**) Family *Lachnospiraceae*. (**D**) Family *Lactobacillaceae*. (**E**) Family *Erysipelotrichaceae*. (**F**) Family *Streptococcaceae*. (**G**) Family *Ruminococcaceae*. (**H**) Family *Muribaculaceae*. (**I**) Family *Ruminococcaceae*. (**J**) Order Bacteroidales. (**K**) *Bacillus coagulans*. Data (*n =* 7–8) are expressed as the mean ± SEM. Statistical significance (*p* < 0.05) is indicated by different letters (e.g., a, b).

**Table 1 nutrients-13-03161-t001:** Compositions of the experimental diets.

Diets	ND	HFD	HFD-ASEC
Protein (%)	14.3	14.1	14.2
Carbohydrates (%)	76.2	40.7	40.6
Fat (%)	9.5	45.2	45.2
kcal/g	3.8	4.8	4.8
**Ingredients (g/kg diet)**			
Casein (≥95%) (Protein from ASEC)	140	175	140 (35)
L-Cystine	1.8	1.8	1.8
Corn starch (Carbohydrates from ASEC)	465.7	289.3	150.7 (138.6)
Dextrinized corn starch	155	96.4	96.4
Sucrose	100	100	100
Lard	0	200	200
Soybean oil (no additives) (Fat from ASEC)	40	40	23.4 (16.6)
Cellulose (Fiber from ASEC)	50	50	44.8 (5.2)
Mineral mix (AIN-93G-MX)	35	35	35
Vitamin mix (AIN-93-VX)	10	10	10
Choline bitartrate (41.1% choline)	2.5	2.5	2.5

HFD: high-fat diet; HFD-ASEC: high-fat diet-adlay seed extrusion-cooked prebiotic.

## Data Availability

Data are contained within the article or Appendix A.

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
