# Peer review of "Synbiotic Intervention with an Adlay-Based Prebiotic and Probiotics Improved Diet-Induced Metabolic Disturbance in Mice by Modulation of the Gut Microbiota"

_nutrients, 2021, doi:10.3390/nu13093161_

Round 1

Reviewer 1 Report

The article "Synbiotic intervention with an adlay-based prebiotic and probiotics improved diet-induced metabolic disturbance in mice by modulation of the gut microbiota" refers to an extremely important problem in obesity, also among children, observed all over the world. It is therefore necessary to learn more about the effects of obesity on all organs and to look for ways to reduce the negative effects of obesity on the whole organism.

The authors of this study presented the effects of pharmacological intervention (through the use of a prebiotic, probiotic and synbiotic) in mice in which obesity was induced by a high fat diet (HFD).

However, the study was carried out on a relatively small group of examined objects, which is a very important factor limiting the inference, and which the authors do not refer to throughout the manuscript. In the discussion, the authors should mention the factors that will be the weak point of the research. When comparing their own research, they should refer to the results of similar research by other authors, taking into account the number of objects tested. The results of the research presented by the authors, although encouraging, should be treated as a pilot and should constitute an introduction to further research on a much larger number of tested objects.

Line 16-17: “Dietary interventions effectively modulate the gut microbiome, resulting in the regulation of obesity and associated comorbidities” - this sentence is too categorical. You can consider changing "resulting" to "may influence the regulation"

Line 43: consider changing "resolve" to "may be helpfull in resolving"

Line 80: The abbreviation ND was used for the first time – it was not explained before

"Results" need to be redrafted. Currently, instead of referring only to the obtained research results, the authors also provide conclusions resulting from the conducted analysis, e.g.

Points 3.1-3.6 sound like conclusions and should be a description of the results of the analysis carried out

Nowhere in the "results" section the authors refer to whether the differences found during the analysis were statistically significant, e.g. lines-153-154: "In particular, the body weight of the ASEC + PRO group had a moderate decrease compared to the ASEC and PRO groups ”- it should be added what is the value of p in this particular comparison. There are many such sentences in the text, it should be analyzed and corrected, not only in the above-mentioned example.

Similarly, at the end of each of paragraphs 3.1-3.6 in the "Results" section (eg lines 170-173: "Taken together, feeding an HFD for 8 weeks successfully induced obesity in mice, and the delayed intervention with ASEC, PRO, or ASEC +) PRO mitigated the increase in the body weight, adipose tissue hypertrophy, and dyslipidemia ”) are a conclusion and not a description of the results obtained. The last sentences of paragraphs 3.1-3.6 should be moved from the "results" section to the "discussion" section.

Figure descriptions: in the figure description, conclusions are given, and there should be a description of the intervention; in addition, under each figure there is a description of the statistical analysis used, which should be in the "Statistical analysis" section. The p-values ​​in the figures in the presented graphs are also missing. Each figure should contain an explanation of the abbreviations used.

Line 258: no explanation for OUT abbreviation.

Line 260: Fig. 5A replace with Figure 5A.

Line 269: "an" should be deleted.

Line 379: The abbreviation NAFLD was used for the first time – it was not explained before

Line 390: to whom / what are the requests?

In the discussion, the authors of the study should pay attention to the limitations of the study (especially the small group of mice tested), which definitely affects the quality of the conclusions drawn. It should be emphasized that the observations from the conducted study are promising, but should be confirmed on a larger number of mice tested.

Author Response

Dear editors and reviewers,

Thank you for your kind assistance in processing our paper entitled ”Synbiotic intervention with an adlay-based prebiotic and probiotics improved diet-induced metabolic disturbance in mice by modulation of the gut microbiota” for Nutrients (nutrients-1365926). We greatly appreciate your consideration of our modified manuscript for publication.

The revision includes a number of positive changes. We revised our manuscript based point-by-point on the comments. To allow the editors and reviewers to easily assess the revised manuscript, all changes were marked up using the “Track Changes” in the resubmitted manuscript. Our responses to the comments are listed below.

Sincerely,

Cheng Huang, Ph.D.

Professor,

Department of Biotechnology and Laboratory Science in Medicine,

National Yang Ming Chiao Tung University, Taipei, Taiwan

Comments:

Reviewer #1

The article "Synbiotic intervention with an adlay-based prebiotic and probiotics improved diet-induced metabolic disturbance in mice by modulation of the gut microbiota" refers to an extremely important problem in obesity, also among children, observed all over the world. It is therefore necessary to learn more about the effects of obesity on all organs and to look for ways to reduce the negative effects of obesity on the whole organism.

The authors of this study presented the effects of pharmacological intervention (through the use of a prebiotic, probiotic and synbiotic) in mice in which obesity was induced by a high fat diet (HFD).

However, the study was carried out on a relatively small group of examined objects, which is a very important factor limiting the inference, and which the authors do not refer to throughout the manuscript. In the discussion, the authors should mention the factors that will be the weak point of the research. When comparing their own research, they should refer to the results of similar research by other authors, taking into account the number of objects tested. The results of the research presented by the authors, although encouraging, should be treated as a pilot and should constitute an introduction to further research on a much larger number of tested objects.

  1. Line 16-17: “Dietary interventions effectively modulate the gut microbiome, resulting in the regulation of obesity and associated comorbidities” - this sentence is too categorical. You can consider changing "resulting" to "may influence the regulation"

RESPONSE: We really appreciate reviewer’s astute suggestion. As suggested, we rephrase the sentence in the corresponding Line 18 in the revised manuscript.

  1. Line 43: consider changing "resolve" to "may be helpfull in resolving"

RESPONSE: We are grateful for the comment. As suggested, we rephrase the sentence in the corresponding Line 51 in the revised manuscript.

  1. Line 80: The abbreviation ND was used for the first time – it was not explained before

RESPONSE: We are grateful for the comment. We include the full name "normal diet" for the abbreviation ND in the corresponding Line 88 in the revised manuscript.

  1. "Results" need to be redrafted. Currently, instead of referring only to the obtained research results, the authors also provide conclusions resulting from the conducted analysis, e.g. Points 3.1-3.6 sound like conclusions and should be a description of the results of the analysis carried out.

RESPONSE: We are grateful for the comment. We rephrase the sentence as suggested. For your reference, the corresponding sentences in the revised manuscript are as follows.

Line 157–159: 3.1 Effects of an ASEC-based synbiotic diet on body weight gain, adipose tissue hypertrophy, and dyslipidemia in HFD-induced obese mice

Line 200–202: 3.2. Effects of an ASEC-based synbiotic diet on glucose tolerance and insulin resistance in HFD-induced obese mice

Line 230–231: 3.3. Effects of an ASEC-based synbiotic diet on hepatic steatosis in HFD-induced obese mice

Line 263–265: 3.4. Effects of an ASEC-based synbiotic diet on proinflammatory factors in the liver and adipose tissues in HFD-induced obese mice

Line 290–292: 3.5. Effects of an ASEC-based synbiotic diet on the diversity of gut microbiota and gut Bacteroidetes in HFD-induced obese mice

Line 331–332: 3.6. Effects of an ASEC-based synbiotic diet on bacterial dysbiosis in HFD-induced obese mice

  1. Nowhere in the "results" section the authors refer to whether the differences found during the analysis were statistically significant, e.g. lines-153-154: "In particular, the body weight of the ASEC + PRO group had a moderate decrease compared to the ASEC and PRO groups ”- it should be added what is the value of p in this particular comparison. There are many such sentences in the text, it should be analyzed and corrected, not only in the above-mentioned example.

RESPONSE: We are grateful for the comment. Following the reviewer’s suggestion, we added p-values in all comparison sentences throughout the revised manuscript.

  1. Similarly, at the end of each of paragraphs 3.1-3.6 in the "Results" section (eg lines 170-173: "Taken together, feeding an HFD for 8 weeks successfully induced obesity in mice, and the delayed intervention with ASEC, PRO, or ASEC +) PRO mitigated the increase in the body weight, adipose tissue hypertrophy, and dyslipidemia ”) are a conclusion and not a description of the results obtained. The last sentences of paragraphs 3.1-3.6 should be moved from the "results" section to the "discussion" section.

RESPONSE: We are grateful for the comment. We move the last sentences of paragraphs 3.1-3.6 to the Discussion section in the corresponding Line 436–448 in the revised manuscript

  1. Figure descriptions: in the figure description, conclusions are given, and there should be a description of the intervention; in addition, under each figure there is a description of the statistical analysis used, which should be in the "Statistical analysis" section. The p-values ​​in the figures in the presented graphs are also missing. Each figure should contain an explanation of the abbreviations used.

RESPONSE: We are grateful for the comment. According to the reviewer’s suggestion, we modified every figure legend and marked up using the “Track Changes” in the resubmitted manuscript. Moreover, p-values are given in the Result section throughout the revised manuscript.

  1. Line 258: no explanation for OUT abbreviation.

RESPONSE: We are grateful for the comment. We include the full name "operational taxonomic unit" in the corresponding Line 293 in the revised manuscript.

  1. Line 260: Fig. 5A replace with Figure 5A.

RESPONSE: We are grateful for the comment. We replace Fig. 5A replace with "Figure 5A" in Line 295 in the revised manuscript.

  1. Line 269: "an" should be deleted.

RESPONSE: We are grateful for the comment. We deleted "an" in Line 305 in the revised manuscript.

  1. Line 379: The abbreviation NAFLD was used for the first time – it was not explained before

RESPONSE: We are grateful for the comment. We include the full name "nonalcoholic fatty liver disease" in the corresponding Line 428–429 in the revised manuscript.

  1. Line 390: to whom / what are the requests?

RESPONSE: We are grateful for the comment. We rephrase the sentence as follows.

Line 431–433: “Therefore, it will be interesting to further investigate the effects of ASEC-based symbiotic on SCFAs concentrations.”

  1. In the discussion, the authors of the study should pay attention to the limitations of the study (especially the small group of mice tested), which definitely affects the quality of the conclusions drawn. It should be emphasized that the observations from the conducted study are promising, but should be confirmed on a larger number of mice tested.

RESPONSE: We are grateful for the comment. Following the reviewer’s suggestion, we added more information in the Discussion and cite additional references. For your reference, the corresponding sentences in the revised manuscript and references are as follows.

Line 448–453: " Rodent models of diet-induced obesity have been widely used to investigate human metabolic perturbations [46]. Concerning that the effects of ASEC-based synbiotic on metabolic health and gut microbiota were based on a sample size of eight per group in a single mouse strain, this research should be regarded as a pilot study that needs to be further validated in experiment of a larger sample size to help extrapolate to clinical therapeutics."

References:

  1. Preguica, I., et al., Diet-induced rodent models of obesity-related metabolic disorders-A guide to a translational perspective. Obes Rev, 2020. 21(12): p. e13081.

Reviewer 2 Report

In this article, Chiou and colleagues have studied the effect of prebiotic (an adlay seed extrusion cooked), probiotics (Lactobacillus paracasei and Bacillus coagulans) and a mixture of both (symbiotic) on metabolic disorders and microbial dysbiosis in mice fed a high-fat diet.

They conclude that all three treatments have improved metabolic disorders, organ dysfunction and other obesity-related pathological indices. Likewise, the research carried out has shown that it improves the dysbiosis characteristic of this animal model of obesity.

The methods used seem to be appropriate and already validated. The numerous methods (the main parameters have been studied) and results make this manuscript a solid and interesting work. The results are well presented and appropriately discussed.

My recommendations would be:

- Several lactobacilli strains have been reported to reduce intestinal permeability in human intestinal epithelial cell lines and murine models. Although the authors explain it very elegantly in their discussion, researchers should determine the gut permeability. Gut permeability is controlled by several specific tight-junction proteins. Among these, ZO-1 (zonula occludens-1) and occludins have been proposed as key markers of tight-junction integrity. Researchers should determine their gene and protein expression to confirm the effects of the three treatments on colonic segment integrity.

- In accordance with the possible increase in intestinal permeability in HFD-fed mice, plasma LPS levels should also be measured.

- The importance of Toll-like receptors (TLRs) during inflammatory responses is well documented. As the authors explain, bacterial endotoxin LPS activates Toll-like receptor 4 (TLR-4), inducing cytokine expression, so it is necessary to study if there are changes in the gene and protein expression of this receptor.

- The authors should add a paragraph indicating the limitations of their study, mainly with respect to the translation of these experimental results into the clinic.

Author Response

Dear editors and reviewers,

Thank you for your kind assistance in processing our paper entitled ”Synbiotic intervention with an adlay-based prebiotic and probiotics improved diet-induced metabolic disturbance in mice by modulation of the gut microbiota” for Nutrients (nutrients-1365926). We greatly appreciate your consideration of our modified manuscript for publication.

The revision includes a number of positive changes. We revised our manuscript based point-by-point on the comments. To allow the editors and reviewers to easily assess the revised manuscript, all changes were marked up using the “Track Changes” in the resubmitted manuscript. Our responses to the comments are listed below.

Sincerely,

Cheng Huang, Ph.D.

Professor,

Department of Biotechnology and Laboratory Science in Medicine,

National Yang Ming Chiao Tung University, Taipei, Taiwan

Comments:

Reviewer #2

In this article, Chiou and colleagues have studied the effect of prebiotic (an adlay seed extrusion cooked), probiotics (Lactobacillus paracasei and Bacillus coagulans) and a mixture of both (symbiotic) on metabolic disorders and microbial dysbiosis in mice fed a high-fat diet.

They conclude that all three treatments have improved metabolic disorders, organ dysfunction and other obesity-related pathological indices. Likewise, the research carried out has shown that it improves the dysbiosis characteristic of this animal model of obesity.

The methods used seem to be appropriate and already validated. The numerous methods (the main parameters have been studied) and results make this manuscript a solid and interesting work. The results are well presented and appropriately discussed.

My recommendations would be:

  1. Several lactobacilli strains have been reported to reduce intestinal permeability in human intestinal epithelial cell lines and murine models. Although the authors explain it very elegantly in their discussion, researchers should determine the gut permeability. Gut permeability is controlled by several specific tight-junction proteins. Among these, ZO-1 (zonula occludens-1) and occludins have been proposed as key markers of tight-junction integrity. Researchers should determine their gene and protein expression to confirm the effects of the three treatments on colonic segment integrity.

RESPONSE: We really appreciate reviewer’s astute suggestion. Following the reviewer’s suggestion, we include the information about the tight-junction integrity in the Discussion and cite additional references. For your reference, the corresponding sentences in the revised manuscript and references are as follows.

Line 409–413: "Moreover, the expression levels of tight junction proteins, including occludins, claudins, and zonula occludens [41], are worth investigating given the close association between the loss of intestinal epithelial barrier integrity and the onset of metabolic disorders to confirm the effects of adlay-based synbiotic treatment on colonic segment integrity."

References:

  1. Chelakkot, C., J. Ghim, and S.H. Ryu, Mechanisms regulating intestinal barrier integrity and its pathological implications. Exp Mol Med, 2018. 50(8): p. 1-9

  1. In accordance with the possible increase in intestinal permeability in HFD-fed mice, plasma LPS levels should also be measured.

RESPONSE: We are grateful for the comment. We added this suggestion in the Discussion section. For your reference, the corresponding sentence in the revised manuscript is as follows.

Line 405–406: "However, this inference has to be further validated by the plasma LPS level and the expression levels of TLR4 and CD14 in intestinal epithelial cells."

  1. The importance of Toll-like receptors (TLRs) during inflammatory responses is well documented. As the authors explain, bacterial endotoxin LPS activates Toll-like receptor 4 (TLR-4), inducing cytokine expression, so it is necessary to study if there are changes in the gene and protein expression of this receptor.

RESPONSE: We are grateful for the comment. We added this suggestion in the Discussion section. For your reference, the corresponding sentence in the revised manuscript is as follows.

Line 405–406: "However, this inference has to be further validated by the plasma LPS level and the expression levels of TLR4 and CD14 in intestinal epithelial cells."

  1. The authors should add a paragraph indicating the limitations of their study, mainly with respect to the translation of these experimental results into the clinic.

RESPONSE: We are grateful for the comment. Following the reviewer’s suggestion, we added more information in the Discussion and cite additional references. For your reference, the corresponding sentences in the revised manuscript and references are as follows.

Line 448–453: " Rodent models of diet-induced obesity have been widely used to investigate human metabolic perturbations [46]. Concerning that the effects of ASEC-based synbiotic on metabolic health and gut microbiota were based on a sample size of eight per group in a single mouse strain, this research should be regarded as a pilot study that needs to be further validated in experiment of a larger sample size to help extrapolate to clinical therapeutics."

References:

  1. Preguica, I., et al., Diet-induced rodent models of obesity-related metabolic disorders-A guide to a translational perspective. Obes Rev, 2020. 21(12): p. e13081.
